# Phase Ib dose-escalation study of the hypoxia-modifier Myo-inositol trispyrophosphate in patients with hepatopancreatobiliary tumors

Marcel A. Schneider [1,2], Michael Linecker[1,2], Ralph Fritsch[1,3], Urs J. Muehlematter [4], Daniel Stocker[4], Bernhard Pestalozzi[1,3], Panagiotis Samaras[5], Alexander Jetter [6], Philipp Kron[1,2], Henrik Petrowsky[1,2], Claude Nicolau[7], Jean-Marie Lehn[8], Bostjan Humar[1,2], Rolf Graf [1,2], Pierre-Alain Clavien[1,2✉] & Perparim Limani [1,2✉]

Hypoxia is prominent in solid tumors and a recognized driver of malignancy. Thus far, targeting tumor hypoxia has remained unsuccessful. Myo-inositol trispyrophosphate (ITPP) is a re-oxygenating compound without apparent toxicity. In preclinical models, ITPP potentiates the efficacy of subsequent chemotherapy through vascular normalization. Here, we report the results of an unrandomized, open-labeled, 3 + 3 dose-escalation phase Ib study (NCT02528526) including 28 patients with advanced primary hepatopancreatobiliary malignancies and liver metastases of colorectal cancer receiving nine 8h-infusions of ITPP over three weeks across eight dose levels (1'866-14'500 mg/m$^2$/dose), followed by standard chemotherapy. Primary objectives are assessment of the safety and tolerability and establishment of the maximum tolerated dose, while secondary objectives include assessment of pharmacokinetics, antitumor activity via radiological evaluation and assessment of circulatory tumor-specific and angiogenic markers. The maximum tolerated dose is 12,390 mg/m$^2$, and ITPP treatment results in 32 treatment-related toxicities (mostly hypercalcemia) that require little or no intervention. 52% of patients have morphological disease stabilization under ITPP monotherapy. Following subsequent chemotherapy, 10% show partial responses while 60% have stable disease. Decreases in angiogenic markers are noted in ~60% of patients after ITPP and tend to correlate with responses and survival after chemotherapy.

[1] Swiss Hepato-Pancreato-Biliary (HPB) and Transplantation Center, University Hospital Zurich, Raemistrasse 100, Zurich, Switzerland. [2] Department of Surgery & Transplantation, University Hospital Zurich, Raemistrasse 100, Zurich, Switzerland. [3] Department of Oncology, University Hospital Zurich, Raemistrasse 100, Zurich, Switzerland. [4] Institute of Interventional and Diagnostic Radiology, University Hospital Zurich, Raemistrasse 100, Zurich, Switzerland. [5] Oncology Center, Hirslanden Hospital Zurich, Witellikerstrasse 40, Zurich, Switzerland. [6] Department of Clinical Pharmacology and Toxicology, University Hospital Zurich, Raemistrasse 100, Zurich, Switzerland. [7] Friedman School of Nutrition Science and Policy, Tufts University, 150 Harrison Ave, Boston, MA, USA. [8] Institut de Science et d'Ingénierie Supramoléculaires (ISIS), Université de Strasbourg, 8 allée Gaspard Monge, Strasbourg, France. These authors share last: Pierre-Alain Clavien & Përparim Limani ✉email: clavien@access.uzh.ch; perparim.limani@usz.ch

Hypoxia occurs in almost all solid tumors and contributes to invasiveness and metastasis, impaired immune responses, and changes in tumor metabolism[1–4]. The lack of oxygen renders tumors resistant to radiotherapy[5] and provokes an angiogenic response resulting in a chaotic, leaky tumor vasculature[6]. The latter hinders efficient delivery of compounds, leading to resistance towards chemo-, immuno- and targeted therapies[7].

Molecularly, hypoxia leads to the stabilization of Hypoxia-Inducible Factors (HIF), key transcription factors that induce gene expression underlying the cellular responses to hypoxia[8]. In tumors, HIF-induced overproduction of angiogenic molecules such as vascular endothelial growth factor alpha (VEGFA) results in the formation of irregular, inefficient vessels[6]. Other HIF-promoted processes include inflammation (e.g., via the NF-κB pathway)[9], metabolic adaptations (e.g., the Warburg effect via up-regulation of glucose transporters such as GLUT1/ SLC2A1)[10], invasiveness (e.g., via Twist, an inducer of the epithelial-mesenchymal transition)[11], stemness (e.g., via OCT4 and other stem cell molecules)[12] and the suppression of adaptive immunity[13,14]—in other words processes that contribute to the progression of malignancy[15]. Not surprisingly therefore, the presence of hypoxia worsens outcome for many tumor types[5,16].

Anti-angiogenic agents targeting hypoxia-induced tumor vasculature have become clinical reality. However, these agents confer only modest survival benefits[17], likely because they can worsen hypoxia, thereby promoting malignant behavior[3,18]. Direct hypoxia-targeting approaches have been only scarcely investigated to date[16]. HIF inhibitors such as PX-478 were tested among multiple cancer types such as colorectal[19] and pancreatic[20,21], but thus far clinical outcomes have been disappointing due to toxicity or lack of effect[22]. Hypoxia-activated prodrugs such as evofosfamide, releasing bromo-isophosphoramide mustard in hypoxic tumor microenvironments, have shown promising results in preclinical, phase I & II studies in pancreatic[23–25], biliary[26], liver[27] and colorectal cancer[28] as well as soft tissue sarcoma[29,30] among other tumor types[31]. However, it failed to show benefits on survival in large scale phase III trials of soft tissue sarcoma[32] and further clinical development was subsequently abandoned. Alternative strategies have aimed at reversing tumor hypoxia per se, however neither blood transfusions[33], nitroglycerin[34], carbogen/nicotinamide[35], nor hyperbaric oxygen[36] have led to the desired effects. In contrast, re-oxygenation of tumors remains the mechanistically simplest yet most holistic approach to counteract the detrimental consequences of hypoxia. Successful re-oxygenation might therefore be superior to existing strategies and likely effective across many cancer types[2,5]. The concept of vessel normalization for the enhancement of standard treatment is therefore of paramount relevance for cancer management.

Myo-inositol trispyrophosphate (ITPP) is a first-of-its-class, anti-hypoxic compound that acts as an allosteric effector of hemoglobin to promote the release of oxygen under conditions of low $pO_2$[37]. In preclinical models, ITPP re-installs tumor normoxia and suppresses the hypoxic response[38–42]. While ITPP can have antitumor activity on its own, its salient property relevant to cancer treatment is the normalization of tumor associated vessels and the subsequent potentiation of chemotherapy effects[38–41]. Importantly, vascular normalization through ITPP appears long-lasting[40,41], suggesting it may create a window of therapeutic opportunity. Moreover, no apparent toxicities were noted in either animals[38–42] or a phase 1a study of healthy volunteers.

Human colorectal and pancreatic ductal adenocarcinoma (PDAC) as well as hepatocellular (HCC) and cholangiocarcinoma (CCA) exhibit hypoxic tumor microenvironments and are associated with intermediate to high hypoxia scores on large scale transcriptional analyses[2]. These tumors therefore quintessentially qualify for the evaluation of anti-hypoxic therapies. The selection of these gastrointestinal tumor entities is furthermore based on availability of promising preclinical efficacy data of ITPP and other anti-hypoxic agents such as evofosfamide obtained with murine colorectal[40–42], hepatoma[43] and pancreatic[39] cancer cell lines and confinement to one anatomical region amenable to reliable radiological tumor assessment.

Here, we report the results of a phase Ib dose escalation study (Fig. 1a) evaluating safety and tolerability of ITPP to define a maximum tolerated dose (MTD), analyze the pharmacokinetics of increasing ITPP doses, and estimating the efficacy in patients with unresectable primary malignancies of the liver, pancreas and biliary tract or liver metastases of colorectal cancer. We show that ITPP is well tolerated up to a MTD of 12,390 mg/m$^2$, with only minimal treatment-associated side effects. Furthermore, ITPP treatment leads to decreases in angiogenic markers which tend to correlate with radiological responses upon subsequent chemotherapy.

## Results

**Baseline characteristics**. 28 patients (18 males and 10 females) with a median age of 65 years (IQR: 53–69) were included in the study between 04/27/2015 to 07/06/2018. Patients were diagnosed with PDAC ($n = 10$), colorectal cancer liver metastases (CRLM, $n = 8$), CCA ($n = 7$), and HCC ($n = 3$). 25/28 of patients had received extensive previous anti-tumor therapies (median of two regimens, IQR 1–4) prior to study inclusion, with a median of two involved organs at study start (IQR 2–3). Details regarding baseline patient characteristics can be found in Supplementary Table 1.

**Dose escalation**. Four patients were included in cohorts 1, 4, 6 & 8, and three in cohorts 2, 3, 5 & 7. 27 patients reached the study endpoint, receiving on average 8.6 of the nine planned ITPP infusions within three weeks as prescribed per protocol. One premature study dropout occurred in cohort 1 due to rapid oncological progression after application of two infusions (data excluded for response and efficacy analyses). No significant treatment-emergent toxicity (sTET) or dose limiting toxicity (DLT) was encountered in cohorts 1–7. A first sTET (Common Terminology Criteria for Adverse Events/CTCAE grade II, hypercalcemia of free ionized calcium) was encountered in the 3rd patient (Nr. 27) of cohort 8 (single dose 14'500 mg/m$^2$). A second sTET and subsequent DLT (CTCAE grade II and IV, both hypercalcemia of free ionized calcium) were encountered in the 4th patient (Nr. 28) of cohort 8. Therefore, the dose of cohort 7 (single dose 12'390 mg/m$^2$) was defined as the MTD of intravenous ITPP administration over eight hours.

**Primary outcome: safety & tolerability**. A total of 56 adverse events (AE) were recorded during ITPP administration and subsequent 10-day follow-up. 24 AE were judged to be related to the underlying medical condition and unlikely to be ITPP-induced. 32 AE were regarded as being at least possibly related to ITPP and hence counted as TET (Table 1). Hypercalcemia of free ionized calcium was the most common TET (67.9% of patients) and was responsible for the sTET and DLT encountered in cohort 8. Hypercalcemia consistently occurred upon ITPP administration start and quickly normalized following cessation of infusions, suggesting it was related to the $CaCl_2$ admixed to minimize ITPP-chelating effects. All patients developing hypercalcemia remained asymptomatic, therefore neither pharmacological intervention nor infusion termination were indicated.

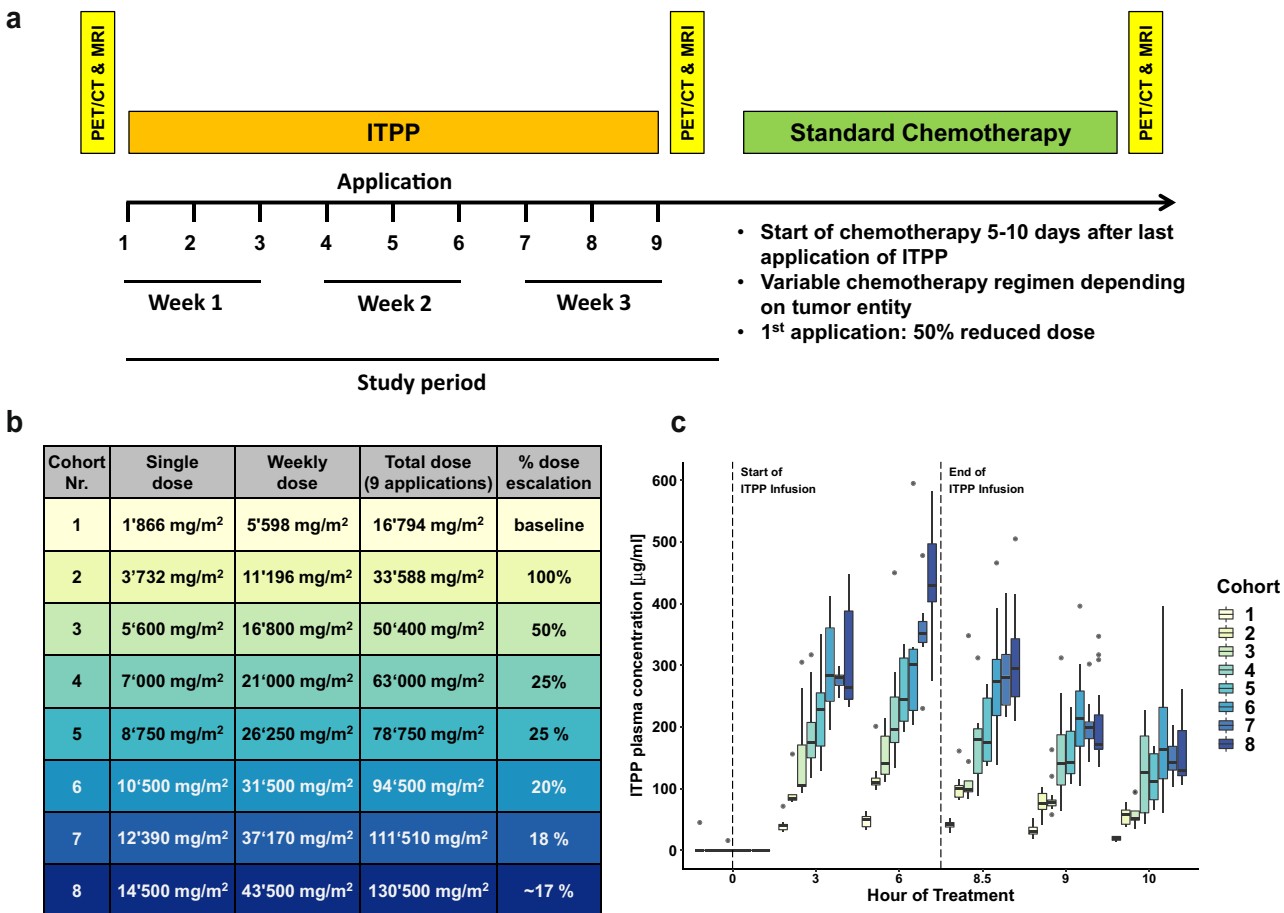

**Fig. 1 Study setup and pharmacokinetics. a** Schema of study flow with timepoints of ITPP administration and assessments. **b** Dose-escalation schema with single, weekly, and total doses for different cohorts. **c** Boxplots displaying median plasma concentrations at start of infusion (hour 0), 3 and 6 h as wells as 30 min, 1 and 2 h after end of intravenous ITPP administration (hour 8) of increasing doses in cohorts on treatment days. Upper and lower ends of boxplots represent 25th and 75th quartiles. Whiskers extend to values within 1.5 * IQR from the boxplot, with data beyond plotted separately as outliers ($n = 3$–4 patients per cohort with 1 measurement performed in technical duplicates per timepoint for each treatment day (normally 9) per patient).

Hypomagnesemia and hypophosphatemia (all CTCAE grade I) were observed in 5/28 and 4/28 patients, respectively. These electrolyte imbalances were asymptomatic and treated by oral supplementation of magnesium or phosphate. One patient developed asymptomatic grade I hyperphosphatemia, which remains potentially related to ITPP administration.

Elevation of blood pressure (<20 mmHg increase in diastolic blood pressure, grade I) occurred in three older male patients with pre-existing hypertension. Following infusion termination, blood pressure levels consistently returned to baseline levels within several hours. All three patients remained asymptomatic and did not require intervention. In summary, ITPP infusions up to the MTD were safe and well tolerated with minimal side effects.

**Secondary outcome: pharmacokinetics.** Increasing doses of the different cohorts (Fig. 1b) resulted in consecutively higher peak plasma levels ($C_{max}$) of ITPP at 6 h after infusion start (Fig. 1c), paralleled by higher circulatory and systemic bioavailability as evidenced by consecutively larger values of area-under-the-curve (AUC) and area-under-the-first-moment-curve (AUMC, Table 2). Following infusion end, plasma concentrations rapidly declined, and higher doses of ITPP did not influence its elimination rate or half-life (1.3–3.3 h for the different cohorts). Furthermore, mean residual time (MRT), clearance, and the volume of distribution at steady state were similar among cohorts. ITPP baseline levels at

infusion start ($C_{min}$) were consistently below the detection threshold, indicating rapid plasma clearance without systemic accumulation (individual patient data in Supplementary Table 2). Therefore, increasing doses of ITPP result in higher systemic exposure with similar drug clearance and elimination.

**Secondary outcome: radiological responses.** Of the 27 patients assessed for efficacy after ITPP monotherapy, 14 patients had morphologically stable disease (SD, median 3d after last ITPP dose), while 11 progressed (PD) according to RECIST1.1 criteria (Fig. 2a). Follow-up imaging after ITPP was unavailable for two patients. Evaluation of metabolic activity (EORTC) yielded four partial metabolic responses (PMR), 11 stable diseases (SMD) and 10 progressions (PMD) (Fig. 2b). Radiological tumor responses after chemotherapy (median 3 cycles) subsequent to ITPP administration (median 94d after last ITPP dose, individual patient data follow up data are provided in Supplementary Table 3) were available for 20 (RECIST: 6 PD, 12 SD and 2 partial response (PR), Fig. 2c) and 15 patients (EORTC: 4 PMD, 7 SMD and 4 PMR, Fig. 2d), respectively.

Cohort/dose-dependent effects were observed for neither ITPP monotherapy nor subsequent chemotherapy. According to RECIST criteria however, median change (MC) in target lesion size appeared better for HCC (0.81% after ITPP, −30.46% after chemotherapy) and CCA (0.88% after ITPP, −18.18% after chemotherapy) than for PDAC (12.16% after ITPP, 0% after

**Table 1 Summary of adverse events and treatment-emergent toxicities by cohort.**

| Cohort (Single dose) | 1 (1866 mg/m²) n = 4 | 2 (3732 mg/m²) n = 3 | 3 (5600 mg/m²) n = 3 | 4 (7000 mg/m²) n = 4 | 5 (8750 mg/m²) n = 3 | 6 (10,500 mg/m²) n = 4 | 7 (12,390 mg/m²) n = 3 | 8 (14,500 mg/m²) n = 4 | All n = 28 |
|---|---|---|---|---|---|---|---|---|---|
| **Treatment-emergent toxicities (TET) judged to be definitively, probably or possibly related to ITPP** | | | | | | | | | |
| Hypercalcemia | 0 | 1x Grade I | 1x Grade I | 4x Grade I | 2x Grade I | 4x Grade I | 3x Grade I | 2x Grade I 1x Grade II (sTET) 1x Grade IV (DLT) | 17x Grade I 1x Grade II 1x Grade IV Total: 19 (67.9%) |
| Hypomagnesemia | 0 | 1x Grade I | 0 | 0 | 0 | 1x Grade I | 0 | 3x Grade I | Total: 5 (17.9%) |
| Hypophosphatemia | 0 | 0 | 1x Grade I | 3x Grade I | 0 | 0 | 0 | 0 | Total: 4 (14.3%) |
| Hyperphosphatemia | 0 | 0 | 0 | 0 | 0 | 0 | 0 | 1x Grade I | Total: 1 (3.6%) |
| Hypertension | 0 | 0 | 0 | 1x Grade I | 0 | 0 | 0 | 2x Grade I | Total: 3 (10.7%) |
| **Other adverse events (AE) & serious adverse events (SAE) judged to be unlikely or definitively unrelated to ITPP** | | | | | | | | | |
| Acute kidney injury grade 3 | 0 | 0 | 0 | 1x Grade III (SAE) | 0 | 0 | 0 | 0 | Total: 1 (3.6%) |
| Ascites due to oncological progress | 1x Grade V (SAE) | 0 | 0 | 1x Grade I | 0 | 0 | 1x Grade III | 0 | Total: 3 (10.7%) |
| AV Block 1° | 0 | 0 | 0 | 0 | 0 | 0 | 0 | 1x Grade I | Total: 1 (3.6%) |
| C. difficile enteritis | 0 | 0 | 1x Grade III (SAE) | 0 | 0 | 0 | 0 | 0 | Total: 1 (3.6%) |
| Cholestasis | 0 | 0 | 1x Grade III (SAE) | 0 | 0 | 0 | 0 | 0 | Total: 1 (3.6%) |
| Fatigue | 0 | 0 | 0 | 1x Grade III (SAE) | 0 | 0 | 0 | 0 | Total: 1 (3.6%) |
| Hypokalemia | 0 | 0 | 0 | 0 | 1x Grade II | 0 | 1x Grade I | 0 | Total: 2 (7.1%) |
| Hyponatremia | 0 | 0 | 0 | 0 | 1x Grade I | 0 | 1x Grade I 1x Grade II | 0 | Total: 3 (10.7%) |
| Icterus due to malignant obstruction | 0 | 0 | 0 | 0 | 1x Grade III | 0 | 0 | 0 | Total: 1 (3.6%) |
| Lower gastrointestinal bleeding | 1x Grade II (SAE) | 0 | 0 | 0 | 0 | 0 | 0 | 0 | Total: 1 (3.6%) |
| Nausea | 0 | 0 | 0 | 1x Grade I | 0 | 0 | 0 | 0 | Total: 1 (3.6%) |
| Nausea & vomiting due to malignant infiltration of stomach | 0 | 1x Grade II (SAE) | 0 | 0 | 0 | 0 | 0 | 0 | Total: 1 (3.6%) |
| Neutropenia | 0 | 0 | 0 | 0 | 0 | 1x Grade III | 0 | 0 | Total: 1 (3.6%) |
| Palpitations | 0 | 0 | 1x Grade I | 0 | 0 | 0 | 0 | 0 | Total: 1 (3.6%) |
| Pyrexia | 1x Grade I | 0 | 0 | 0 | 0 | 0 | 1x Grade I | 0 | Total: 2 (7.1%) |
| Upper respiratory tract infection | 0 | 0 | 0 | 2x Grade I | 0 | 0 | 1x Grade II | 0 | Total: 3 (10.7%) |

AE adverse event, SAE serious adverse event, sTET significant treatment-emergent toxicities, DLT dose-limiting toxicity. AE were graded as SAE if resulting in death, immediately life-threatening, necessitating hospitalization or resulting in persistent health damage

**Table 2 Pharmacokinetics of intravenous ITPP administration per cohort.**

| Cohort (Single dose) | $C_{min}$ [mg/L] | $C_{max}$ [mg/L] | $T_{max}$ [h] | $T_{last}$ [h] | $C_{last}$ [mg/L] | Half life/ $T_{0.5}$ [h] | Elimination rate $\lambda_z$ L/h] | AUC [h*mg/L] | AUMC [h2*mg/L] | MRT [h] | CL [L/h] | VSS [L] |
|---|---|---|---|---|---|---|---|---|---|---|---|---|
| 1 (1866 mg/m2) | 0.0 | 47.2 | 6.0 | 10.0 | 19.2 | 1.3 | 0.5 | 350.7 | 1994.7 | 1.7 | 4.8 | 11.0 |
| 2 (3732 mg/m2) | 0.0 | 120.6 | 6.0 | 10.0 | 55.8 | 1.8 | 0.4 | 847.4 | 4925.6 | 1.8 | 3.8 | 10.5 |
| 3 (5600 mg/m2) | 0.0 | 153.4 | 6.0 | 10.0 | 59.5 | 1.4 | 0.5 | 1168.7 | 6445.9 | 1.5 | 4.4 | 9.2 |
| 4 (7000 mg/m2) | 0.0 | 228.0 | 6.0 | 10.0 | 127.5 | 3.3 | 0.2 | 1649.5 | 9421.8 | 1.7 | 3.1 | 12.8 |
| 5 (8750 mg/m2) | 0.0 | 257.8 | 6.0 | 10.0 | 116.8 | 2.1 | 0.3 | 1850.5 | 10396.4 | 1.6 | 4.0 | 11.1 |
| 6 (10500 mg/m2) | 0.0 | 313.4 | 6.0 | 10.0 | 186.4 | 2.6 | 0.3 | 2439.2 | 13886.7 | 1.7 | 3.3 | 11.7 |
| 7 (12390 mg/m2) | 0.0 | 353.6 | 6.0 | 10.0 | 149.4 | 1.7 | 0.4 | 2457.2 | 14095.2 | 1.7 | 4.4 | 11.4 |
| 8 (14500 mg/m2) | 0.0 | 439.2 | 6.0 | 10.0 | 166.0 | 1.8 | 0.4 | 2864.1 | 16280.4 | 1.7 | 4.4 | 11.4 |

$C_{min}$ [mg/L] = minimal concentration at start of infusion; $C_{min}$ [mg/L] = maximum concentration; $C_{max}$ [mg/L] = maximum concentration; $T_{max}$ [h] = time of maximum concentration; $T_{last}$ [h] = time of last positive concentration observed; $T_{last}$ [h] = last positive concentration; Half-life / $T_{0.5}$ [h] = half-life by lambda z; Elimination rate $\lambda_z$ [L/h] = lambda z, negative of best fit terminal slope; AUC [h*mg/L] = Area under the curve from 0 to $T_{last}$; AUMC [h2*mg/L] = Area under the first moment curve to the $T_{last}$; MRT [h] = mean residence time; CL [L/h] = clearance; VSS [L] = volume of distribution at steady state.

chemotherapy) and CRLM (6.67% after ITPP, −3.12% after chemotherapy). According to EORTC criteria, PDAC showed decreases in metabolic activity under ITPP monotherapy (MC: −3.36%), while other tumor types increased (CCA: 17.97%, CRLM: 40.1%, HCC: 15.54%). Following chemotherapy, however, all tumor types displayed decreased metabolic activity (CCA: −32.39%, CRLM: −14.64%. HCC: −8.47%, PDAC: −52.19%, Supplementary Fig. 1).

**Secondary outcome: biochemical serum responses**. Circulatory tumor markers including carcinoembryonic antigen (CEA) for CRLM, α-fetoprotein (AFP) for HCC, and carbohydrate antigen 19-9 (CA19-9) for PDAC and CCA were not elevated in 5/27 patients. Of the remaining 22, 11 patients showed a decrease, while 11 experienced an increase in marker levels. However, no clear correlation to tumor type or dose administered could be distinguished. Furthermore, changes in tumor markers correlated neither to radiological responses after ITPP or chemotherapy, nor to survival (Supplementary Fig. 2).

To examine the effects of ITPP on tumor-associated angiogenesis, five prominent circulatory pro-angiogenic factors (VEGFA, ANG1/2, EGF, PECAM1/CD31) were assessed during ITPP monotherapy. Following ITPP treatment, VEGFA was reduced in 44.4% (12/27), ANG1 in 51.9% (14/27), ANG2 in 59.3% (16/27), EGF in 85.2% (23/27), and PECAM1 in 66.7% (18/27) of patients (Fig. 3). No correlation was evident to cohorts nor to radiological responses after ITPP monotherapy. We also found no consistent changes by tumor type, although overall decreases in angiogenic markers were most prominent for PDAC, followed by CCA (Supplementary Fig. 3). In contrast, patients with decreased angiogenic markers tended towards better radiological responses following subsequent chemotherapy. Moreover, angiogenic reductions tended to correlate with improved survival after chemotherapy, with patients experiencing a lowering of VEGFA (398 vs. 196 days, $p = 0.053$) and PECAM1/CD31 (380 vs. 171 days, $p = 0.36$) benefiting of a two-fold longer overall survival (Fig. 3). The angiogenic molecules displayed significant inter-marker correlations but no correlation to tumor-specific markers (Supplementary Fig. 4). All angiogenic markers tended to display inverse correlations with patient overall survival (OS), particularly PECAM1 ($R = −0.46$, $p = 0.015$), supporting an association between reduced angiogenic activity after ITPP and an improved survival after subsequent chemotherapy.

**Ad hoc outcome: tissue marker responses on anti-hypoxic therapy in one patient**. Regular tissue biopsies before and after ITPP treatment for assessment of changes in hypoxia-mediated gene expression, although possibly providing meaningful insight into the anti-hypoxic effects of ITPP, were precluded by the responsible ethic committee due to safety concerns in this phase I trial. However, patient nr. 21 (61-year-old female suffering of CRLM) initially underwent left hemi-hepatectomy for liver metastases before being included in the trial. After ITPP treatment and 2 months of subsequent chemotherapy, the patient showed radiological stable disease and followingly underwent open surgical microwave ablation of persisting liver metastases based on the recommendation of the interdisciplinary tumor board. Tissue biopsies harboring tumor cells invading into liver parenchyma of both interventions were obtained and expression of hypoxia tissue markers compared by immunohistochemistry. Cancer cells were distinguished from stromal cells of the tumor microenvironment and hepatocytes by hematoxylin eosin, Masson's trichrome and cytokeratin B staining (Fig. 4a). We found only weak staining of HIF1α and HIF2α, with no difference between before and after ITPP treatment. In contrast, expression

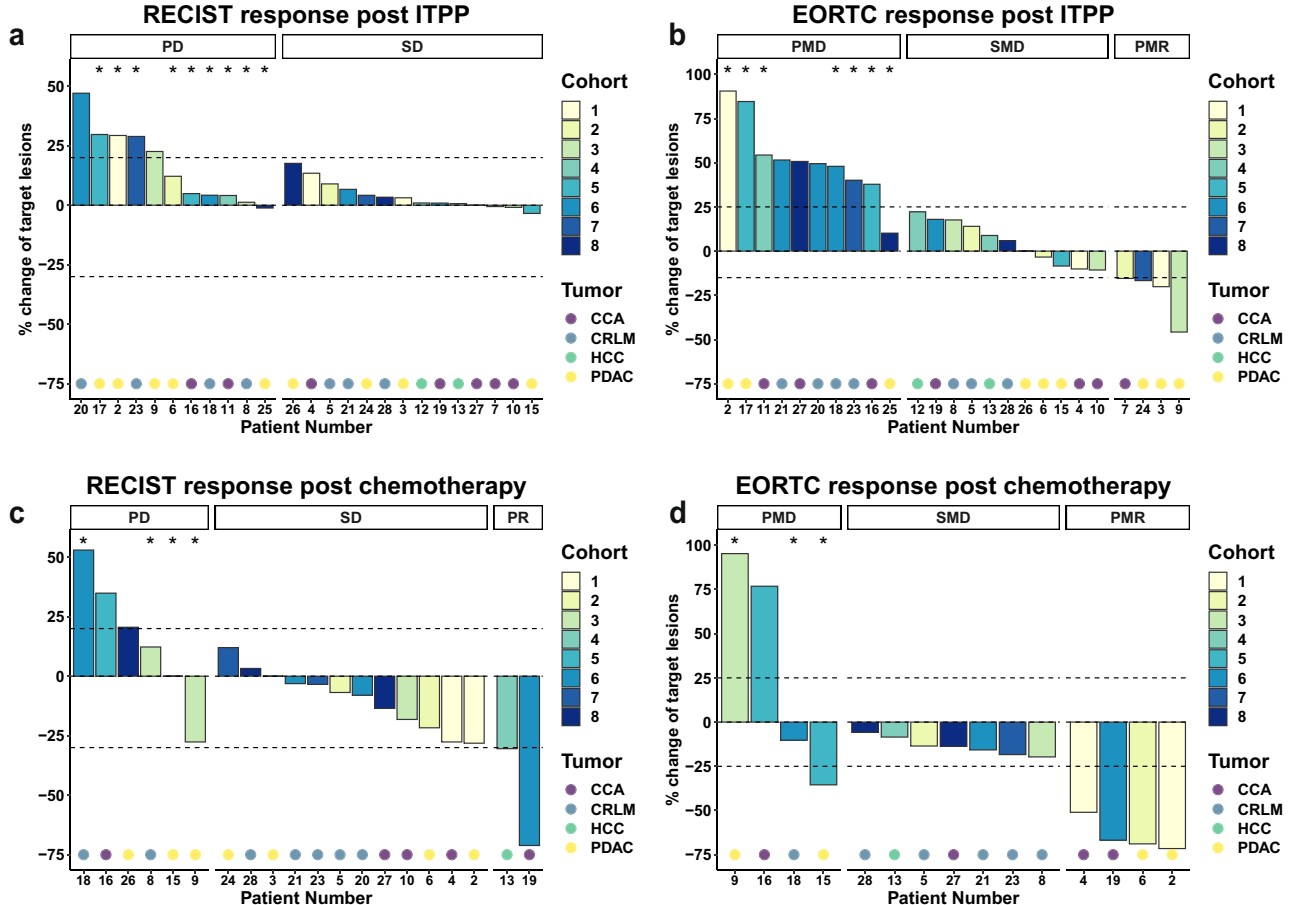

**Fig. 2 Radiological responses post ITPP and chemotherapy.** Displayed as waterfall plots of percental changes of either diameter in millimeters for RECIST1.1 criteria or SUV uptake for EORTC criteria. Morphological changes according to RECIST1.1 criteria (**a**) and metabolic changes according to EORTC criteria (**b**) in target lesions after ITPP monotherapy. Morphological changes according to RECIST1.1 criteria (**c**) and metabolic changes according to EORTC criteria (**d**) in target lesions after subsequent chemotherapy. *Indicates the appearance of new (FDG avid) lesions.

of carbonic anhydrase 9 (CA9), a prominent enzyme transcriptionally regulated through hypoxia responsive elements and a marker of tumor hypoxia[44] contributing to increased tumor progression, acidification, and metastases[45], was decreased after ITPP treatment. Similarly, expression of SLC2A1/GLUT1 observed in cancer cells, which is induced by hypoxia and HIF1α mediating the switch from oxidative phosphorylation to glycolysis[10] and increased expression associated with decreased survival[46], was downregulated after ITPP treatment. Vimentin as a marker of epithelial to mesenchymal transformation (EMT) was decreased after ITPP administration, similar to findings in our preclinical studies[40]. Regarding vasculature, we found similar areas of PECAM1/CD31 positive vessels in tumors before and after ITPP treatment. However, the transcription factor ERG regulating vascular stability and integrity[47] was more abundant after ITPP treatment, suggesting that anti-hypoxic treatment by ITPP might deter the formation of leaky tumor vasculature.

**Ad hoc outcome: survival.** Median progression free survival (PFS) was 48 days for the whole patient population, with no significant differences among tumor types (CCA: 158 days, CRLM: 23 days, HCC: 345 days, PDAC 32 days) or cohorts. Median OS was 206 days from start of ITPP infusions, again with no significant differences among tumor types (CCA: 302 days, CRLM: 340 days, HCC: 684 days, PDAC 165 days) or different doses of ITPP.

## Discussion

This is the first-in-patient report of treatment with the anti-hypoxic compound ITPP, the first-of-its-class anti-hypoxic molecule without toxic effects[37–41,48]. We identified the MTD of ITPP at 12,390 mg/m², confirming high tolerability in patients with advanced tumor burden. 57% of AE were at least possibly related to ITPP and were mostly electrolyte disturbances usually in the form of hypercalcemia (59.4%). Because ITPP is a Ca$^{++}$-chelator, the drug was balanced with CaCl$_2$ for intravenous administration, providing a plausible explanation for hypercalcemia. Hypomagnesemia/-phosphatemia (28.1%) and mild aggravation of hypertension ($n = 3$) were the other TETs occurring at least twice and possibly related to the intravascular volume increase (1 l/8 h) upon ITPP infusion. Importantly, patients with these TETs remained asymptomatic and required no or minimal intervention. Plasma ITPP levels rose with increasing doses and rapidly returned to baseline after infusion, suggesting efficient systemic clearance amenable to subsequent therapies.

This study included a heterogeneous population of patients, most of whom had already undergone multiple treatments, consistent with advanced disease and resistance issues. Overall, ITPP monotherapy was associated with radiological disease stabilization only weakly (52%SD; 41%SMD/15%PMR), while subsequent chemotherapy strengthened these associations and showed efficacy in a meaningful proportion (60%SD/10%PR; 47% SMD/27%PMR). We found no clear correlations of responses with dosing or tumor type, raising the question if the observed

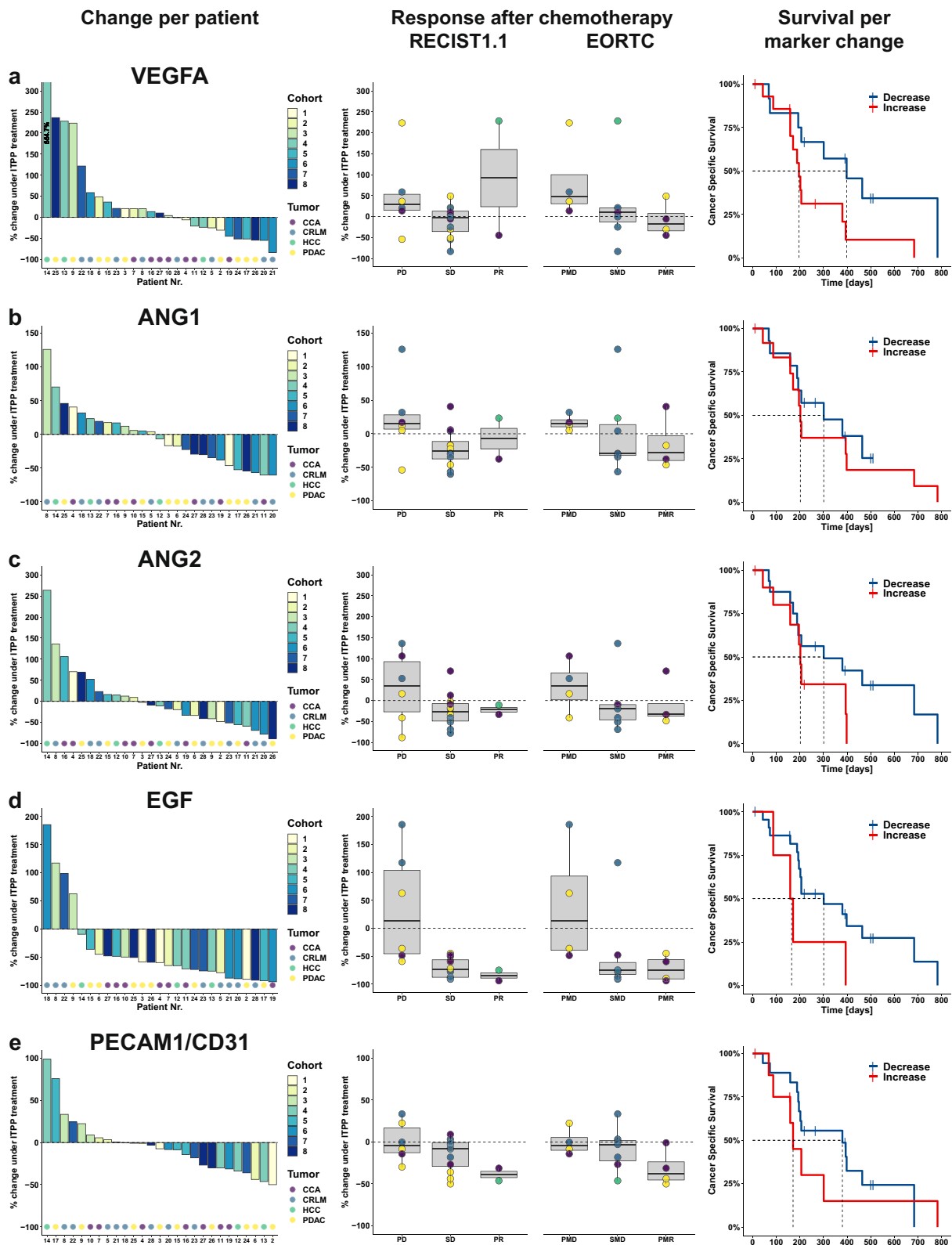

**Fig. 3 Changes in serum angiogenic markers post ITPP and correlation with chemotherapy responses and survival.** Changes in circulating levels of VEGFA (**a**), ANG1 (**b**), ANG2 (**c**), EGF (**d**), and PECAM1/CD31 (**e**). Percental changes of pre- versus post-ITPP monotherapy markers levels are depicted (from left to right) by patient as waterfall plots. Association of morphological and metabolic response post chemotherapy with changes in serum angiogenesis markers under ITPP treatment are shown as boxplots displaying median values with the upper and lower ends representing the 25th and 75th quartiles, respectively. Whiskers extend to values within 1.5 * IQR from the boxplot, with all individual data points shown overlaid and colored according to tumor type. Survival stratified by decreased or increased marker levels displayed as Kaplan–Meier curves. $n = 27$ individual patients, marker measurements performed as technical duplicates.

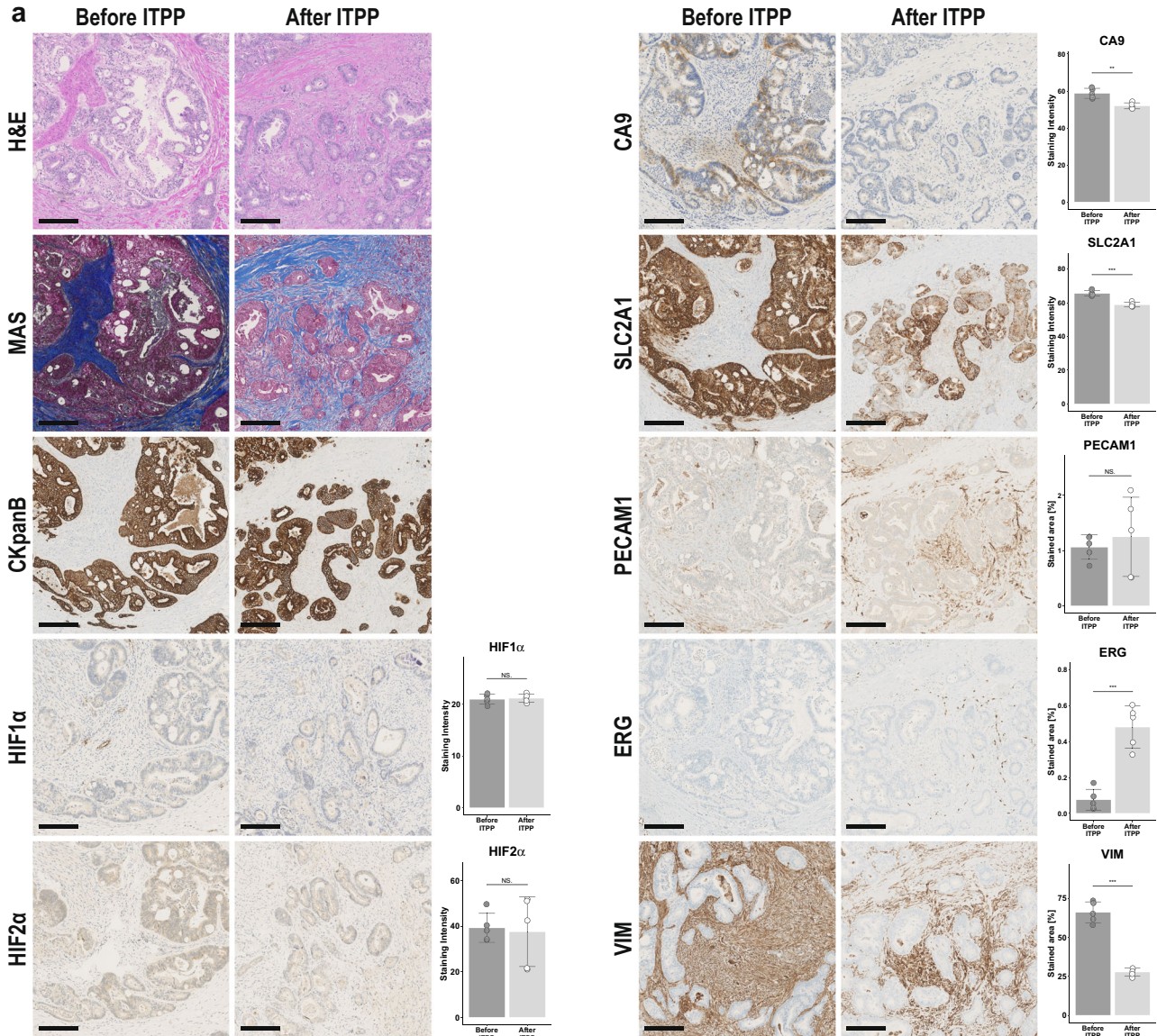

**Fig. 4 Tissue-based hypoxia response markers of one exemplary patient. a** Histological images taken at 20x magnification of the sample taken before ITPP treatment (left hemi-hepatectomy) compared to the sample obtained after ITPP (microwave ablation). Note the decreased expression of hypoxia-mediated genes such as CA9 and SLC2A1 after ITPP treatment and signs of increased vessel maturity by increased EGR expression. Scale bare: 200 µm. *N* = 1 patient. Histological quantifications are shown as boxplots displaying mean values ± standard deviation and overlaid single measurements. Statistical differences derived from two-sided students *t*-test with no adjustment for multiple comparisons. **$p < 0.01$, ***$p \leq 0.001$, NS. = No significant difference.

disease stabilizations might be due to the natural course of the disease or related to the underlying extent of hypoxia in tumors. Pretreatment assessment of tumor hypoxia and its changes following ITPP treatment, e.g., by invasive tumor biopsies or specialized imaging of hypoxia using 18F-Fluoromisonidazole positron emission tomography would have been desirable but were precluded by the responsible ethics committee. These investigations should be implemented in further studies examining the effects of ITPP to assess tumor hypoxia in vivo at baseline and under treatment. To understand whether such improvements were due to ITPP or rather by chance, we therefore assessed surrogate markers related to tumor activity and angiogenesis. Tumor-specific marker (CEA/AFP/CA19-9) responses were mixed and correlated to radiological responses after neither ITPP nor chemotherapy. Therefore, tumor-specific markers either do not reflect tumor mass/activity in our cohort,

or the ITPP effects, if any, on tumor mass and activity are of lesser relevance for potential ITPP benefits.

Importantly, however, ITPP monotherapy led to decreases in angiogenic markers in the majority of patients, reductions that tended to correlate with responses and survival after subsequent chemotherapy. Besides VEGFA, the classic promoter of tumor-associated angiogenesis[49], and ANG1/2, which contribute to tumor angiogenesis in conjunction with VEGFA[50], we assessed EGF, increasingly recognized as a stimulator of hypoxia-mediated angiogenesis[51], and PECAM1/CD31, a read-out for systemic angiogenic activity[52,53]. Overall changes were consistent with an inhibition of angiogenesis by ITPP in around 60% of patients, as evidenced by a concerted downregulation of angiogenic molecules suggested by intermarker correlation, the association between angiogenic marker reductions and stable/regressive disease in patients, and the observed tendency of prolonged survival in

patients with angiogenic reductions. Moreover, angiogenic markers correlated neither to tumor markers nor to radiological responses after ITPP monotherapy.

With no regular tumor biopsies before and after treatment available, we nevertheless assessed tissues available before and after ITPP treatment of one patient. In accordance with the changes observed in serum, we found decreased expression of markers mirroring actual intratumoral hypoxia such as CA9 or SLC2A1/GLUT1 and evidence of increased vessel stability. Although these exemplary findings need to be regarded with caution due to the risk of pure sampling bias in just one patient, the tissue-based expression of hypoxia markers could indicate an actual anti-hypoxic effect of ITPP on tumor cells. Altogether, these findings imply that ITPP may counteract angiogenesis in cancer patients, which in turn may improve outcomes after subsequent chemotherapy. This view is fully consistent with the proposed working mode of ITPP, that is to normalize tumor vessels as to improve the delivery of subsequent chemotherapy[38–41].

Our trial was designed to assess safety and tolerability of ITPP—any conclusions about efficacy hence are preliminary at present. High patient heterogeneity in terms of tumor type, stage, treatment history and small sample size likely obscured responses to ITPP dosing and to tumor type. Several patients did not complete subsequent chemotherapy, further limiting conclusiveness. Alternatively, already small doses of ITPP might suffice for improved oxygenation, with higher doses adding little benefit. Furthermore, our trial is limited by the lack of direct evidence for anti-hypoxic action. The impact of ITPP on vessel normalization and hypoxia regulated target genes needs future confirmation of the presumed mechanism underlying the ITPP benefits. Finally, the reported PFS and OS are of a small and heterogenous phase 1b patient population and should therefore not be overstated. Future trials need to address these open questions. The first step will be a sufficiently powered phase 2 trial comparing standard chemotherapy with and without preceding ITPP treatment in a homogenous group of cancer patients. For now, such a trial should allow for sufficient time between ITPP treatment and subsequent chemotherapy to avoid undesired interaction, and for sufficiently tight ITPP administrations to avoid states of intermittent hypoxia[54]. If tumor vessel normalization, as suggested by our preliminary data, can be confirmed, ITPP might improve the delivery of a range of anticancer compounds, such as oncogene-targeting antibodies (e.g., Cetuximab blocking EGFR may further synergize with the EGF-lowering effects of ITPP) or inhibitors of immune checkpoints. Our data indicate that ITPP on its own is very well tolerated in cancer patients and not associated with toxicity as seen after cytotoxic or targeted treatments including alternate anti-hypoxic approaches such as HIF1α inhibitors[22]. On the contrary, some of our patients reported revitalizing effects upon ITPP treatment, consistent with animal studies demonstrating increased performance capacity through ITPP[48]. Therefore, ITPP fully deserves exploration to further its clinical development.

In conclusion, this phase 1b trial suggests ITPP is tolerated well in patients with advanced cancer. The recommended MTD is 12'390 mg/m$^2$ infused nine times over three weeks. ITPP monotherapy seems to exert antiangiogenic activity that might translate into improved responses towards conventional chemotherapy. These observations remain to be confirmed in further trials. Efforts for a phase 2 dose-extension are currently ongoing.

## Methods

**Study design**. This first-in-patient investigator-initiated phase Ib dose-escalation study followed a 3 + 3 dose-escalation scheme with a planned maximal inclusion of 48 participants allocated to eight cohorts of increasing doses. Outcomes were

safety, tolerability, pharmacokinetics, and preliminary efficacy of ITPP with participants in this exploratory, prospective, open-label and unblinded, unrandomized, single-center investigation being enrolled at the University Hospital Zurich (USZ), Switzerland. Written informed consent was obtained of all participants before study inclusion after a time of consideration of at least 24 h. Written consent included the reporting and publication of individual, anonymized patient data. The study adhered to the principles of the Declaration of Helsinki, current good clinical practice guidelines and all ethical regulations. The study protocol was approved by the responsible independent ethics committee of Zurich (KEK-ZH-Nr. 2014-0374) and the national regulatory authority Swissmedic (2015DR1009) and is provided in the Supplementary Note 1. The study setup was published before inclusion start[55] and the trial first registered at ClinicalTrials.gov (NCT02528526) on 11/11/2014. After 6 sequential reviews requiring minor editorial changes, final registration at ClinicalTrials.gov was obtained on the 08/18/2015. The first patient was included in the study 04/13/2015, with treatment started on the 04/27/2015. 4 patients were included in the trial before final registration at ClinicalTrials.gov (04/13/2015–07/20/2015). The last patient was included on the 06/12/2018, treatment started on the 06/18/2018 and the last treatment administered on 07/06/2018. Data cut-off for the study was the 12/31/2018. The manuscript was written in compliance with ICMJE guidelines.

**Participant eligibility**. Patients were eligible if they were aged ≥18 years and diagnosed with irresectable HPB tumors, including CRLM, PDAC, HCC and CCA, had an Eastern Cooperative Oncology Group performance status score ≤1, and had adequate hematological, renal, and hepatic function (Supplementary Note 1: page 12). Patients were required to have had at least 28 days of recovery from recent surgery or chemo- or radiotherapy.

**Objectives and outcomes**. Primary objectives were (i) assessment of the safety and tolerability of increasing doses of ITPP, and (ii) establishment of the MTD (primary endpoint) according to the dose escalation schema. The primary outcome was measured by collection of adverse effects information according to Common Terminology Criteria for Adverse Events (CTCAE, US National Cancer Institute, version 4.03). Safety objectives and outcomes were identical with the primary endpoint.

Secondary objectives included assessment of (a) pharmacokinetics (outcome measured using repeated blood measurements), and efficacy of ITPP monotherapy by (b) anti-tumor activity (outcome: radiological assessment through magnetic resonance imaging (MRI) and 18-fluorodeoxyglucose positron-emission-tomography (FDG-PET)), and (c) changes of circulatory tumor-specific and angiogenic markers (outcome measured using blood samples before and after ITPP treatment).

Ad hoc outcomes consisted of assessment of overall survival of patients in regard to tumor type and cohort as well as histological assessment of hypoxia markers in tumor tissue of selected patients.

**Study procedures**. The study drug (brand name: OXY111A) was intravenously administered in 9 infusions, each lasting 8 h, over 3 weeks in an outpatient setting at the Phase 1 unit of the USZ Clinical Trials Center. Due to its anionic properties, ITPP acts a potent chelator of calcium. Therefore, administration with CaCl$_2$ (Baxter) at a 1:0.75 molar ratio has been determined to prevent hypocalcaemia. Respective doses were calculated based on body surface area (DuBois/DuBois formula). Cohort 1 started at weekly doses of 5600 mg/m$^2$ weekly, equal to the maximal weekly dose tested in healthy volunteers (unpublished data from Normoxys®). Dose escalation was performed according to the predefined scheme (Fig. 1a, Supplementary Note 1: page 6) up to a maximum weekly dose of 43,700 mg/m$^2$. During infusion days, participants had continuous monitoring of vital signs and daily 12-electrode electrocardiograms. Electrolytes measurements and venous blood gas analyses were performed three-hourly, while complete blood count, coagulation and routine kidney/liver parameters were assessed twice daily.

Chemotherapy was started within 5–10 days after the last ITPP infusion according to the recommendations of the multidisciplinary tumor board. For the first cycle, a 50%-reduced dose was applied to minimize potential interactions of ITPP and conventional cytotoxic agents.

**Safety assessment**. Severity of encountered AE was assessed following the CTCAE guidelines. Serious AE were defined as events being life-threatening, necessitating hospitalization, resulting in death or birth defects. DLT was defined as any AE ≥ grade 3 and sTET as any grade 2 AE considered to be definitely, probably or possibly related to ITPP. Occurrence of 1-2 sTET resulted in three additional patients receiving the same dose. MTD was defined as the dose preceding the level at which 1 patient experienced a DLT or ≥3 patients experienced a sTET (Supplementary Note 1: page 6). The window for DLT/sTET/MTD assessment was from first dose of ITPP until first dose of chemotherapy or 10d following the last ITPP infusion. Dose escalation proceeded when ≥3 patients/cohort had reached the study end and completed final DLT assessment. An interval of missed appointments of >7 days between two ITPP applications or an overall of <5 ITPP applications within 5 weeks resulted in the discontinuation of the participant with a replacement by another patient in the same cohort.

**Radiological evaluations**. Radiological assessment, consisting of abdominal MRI and FDG-PET-CT, was performed before ITPP administration and after the 3-week treatment course. Where possible, radiological re-evaluation was performed after 3–6 cycles of chemotherapy. Radiological responses were evaluated according to the RECIST 1.1 criteria[56] for MRI and EORTC criteria[57] for FDG-PET-CT imaging.

**Biochemical evaluations**. Plasma levels and pharmacokinetics of intravenous ITPP administration at time points 0 h, 3 h, 6 h, 8.5 h (30 min after end of infusion), 9 h and 10 h were measured by SYNLAB Analytics (Birsfelden, Switzerland), and tumor-specific markers by the USZ clinical chemistry department. Serum angiogenesis markers were quantified by multiplex bead-based immunoassays following provided instructions (LEGENDplex™, Biolegend, San Diego, CA/USA). For biochemical markers, the percentage change comparing *before* (day 1 0 h) with *after* (last day 9 h) ITPP treatment was calculated.

**Histological evaluations**. Samples were collected in 4% buffered formalin, dehydrated, embedded in paraffin and cut into 5 μm sections and stained after antigen retrieval. Hematoxylin/eosin and Masson's trichrome stains were performed according to standard protocols. The following antibodies were used for immunohistochemical staining: pan-B cytokeratin (CkpanB, Dako, M3515, dilution 1/50), HIF1α (Abcam, ab16066, dilution 1/400), HIF2α (Abcam, ab199, dilution 1/50), carbonic anhydrase 9 (CA9, Abcam, ab15086, dilution 1/3000), glucose transporter 1/SLC2A1 (GLUT1, Millipore, 07-1401, dilution 1/1000), PECAM1 / CD31 (Dako, M0823, dilution 1/10), ETS transcription factor (ERG, Roche, 790-4576, prediluted) and vimentin (VIM, Dako, M7020, dilution 1/250). For histological analyses, 5 random images of tumor areas of stained slides were taken at 40x magnification. Stained areas were isolated by color deconvolution and thresholding. Quantification of staining intensity was measured on thresholded areas, converted onto a scale from 0 (white, no staining) to 100 (black, completely stained) and compared between the sample taken before ITPP treatment and the one after. For calculation of percentage area stained, thresholded areas were compared to the complete area of the picture. ImageJ (V1.53c, National Institutes of Health, USA) was used for all histological analyses[58].

**Data handling and statistical analyses**. The study, including patient recruitment and accuracy of data collection, was continuously monitored by uninvolved clinical trial managers of the clinical trials center of the university of Zurich. The study was audited twice by external reviewers during the phase of patient recruitment, which objected no relevant irregularities. secuTrial® (V4.9.1.14, Berlin, Germany; licensed by the clinical trial center of the university of Zurich) was used for protected, monitored and version-controlled data capturing during the clinical trial. Microsoft® Excel® (Microsoft 365 Enterprise, Redmond, Washington/US) was used for data export of secuTrial® and storage for subsequent analysis. R V 4.0.2 and R-Studio V1.3.1093 were used for statistical analyses, calculations, and graphical representations. All data (Supplementary Dataset: Source data) and code (Supplementary Software: Source code R markdown file) used for analyses are available in the supplements linked to this article.

Cohort size was based on the traditional 3 + 3 dose-escalation scheme without formal power or sample size calculation. Data are summarized with descriptive statistics using medians and interquartile ranges (IQRs) as indicated. Pearson's coefficient was used to test for correlation between numerical variables, and Mann–Whitney U or Kruskal–Wallis tests for differences in continuous variables among groups. Kaplan–Meier curves and the Mantel–Cox log rank test were used for survival analyses. No adjustment for multiple testing was performed.

**Prior presentation**. Presented in part orally at the 13[th] biennial IHBPA World Congress, Geneva, Switzerland, September 4–7, 2018, the 106[th] annual congress of the Swiss Society of Surgery, Berne, Switzerland, May 15–17, 2019 and as poster at the ESMO Immuno-Oncology Congress 2018, Geneva, Switzerland, December 13–16, 2018.

**Reporting summary**. Further information on research design is available in the Nature Research Reporting Summary linked to this article.

## Data availability

All data underlying the calculations and figures in the study are available in the "Supplementary Dataset: Source data" file linked to this article. Extended anonymized patient baseline, follow up, and pharmacokinetic information are also available in Supplementary Tables 1–3. Detailed patient-related study raw data (e.g., radiological imaging, laboratory value reports, medical letters, etc. containing patient identifiers such as names, date of birth, addresses or affiliated institutions) which could compromise protection of privacy of research participants are not publicly available due to privacy restrictions. These data are available in anonymized form from the corresponding authors (P.-A.C. or P.L.) upon reasonable request. Source data are provided with this paper.

## Code availability

All statistical codes used for analyses in *R* are available in the "Supplementary Software: Source code" *R* markdown file linked to this article. The accompanying information file on the Source code explains the necessary setup for analyses in *R*. Combined with the source data file, this allows reproduction of all calculations and figures.

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

## Acknowledgements

We thank the patients, their families, and caregivers for participation in the trial and the study nurses Karin Pettersson, Livia Löffler, Christiane Nilles, and Astrid Hirt for their instrumental support and caring service towards the patients during study drug application. Furthermore, we thank Lilian Roth, Philipp Dutkowski, Beat Müllhaupt, and Alexander Siebenhüner for their help in patient recruitment and Anurag Gupta and Katherine Hagedorn for their support with the flow cytometric multiplex assay. Finally, we would like to express our gratitude towards the monitoring team of the Clinical Trials Center of Zürich and the whole Phase-1-Unit team of the University Hospital Zürich for their help and cooperation. Pierre-Alain Clavien served as sponsor of the trial. This study was supported by the Clinical Research Priority Program ("non-resectable liver tumors—from palliation to cure") of the University Hospital Zurich, the Sassella Foundation (grants Nr. 12/05 and 13/01), the Candoc Forschungskredit (University of Zurich, FK-13-030), the Liver and Gastrointestinal Disease Foundation, and the Swiss Cancer League (KFS-3262-08-2013).

## Author contributions

Conception and design: P.L., B.P., P.S., C.N., J.-M.L., A.J., H.P., M.A.S., M.L., P.K., B.H., R.G. and P.-A.C. Patient management: M.A.S., M.L., P.K., R.F., P.L., H.P., B.P., P.S. and P.-A.C. Administrative, technical, or material support (organizing trial medication): P.L., R.G., J.-M.L., C.N. and A.J. Acquisition of data: M.A.S., M.L., P.K., P.L., D.S. and U.J.M. Data analysis and interpretation: M.A.S., B.H., R.F., D.S., U.J.M., P.L., C.N., J.-M.L., R.G. and P.-A.C. Manuscript writing: M.A.S., P.L., B.H. and P.-A.C. Manuscript revision: M. A.S., P.L., B.H., P.S., R.G. and P.-A.C. Manuscript approval: all authors. Study supervision: B.H., R.G. and P.-A.C.

## Ethics approval and consent to participate

The study adhered to and was conducted according to the principles of the Declaration of Helsinki and current good clinical practice guidelines. The protocol was approved by the responsible ethics committee of the canton of Zurich, Switzerland (KEK-ZH-Nr. 2014-0374) and the national regulatory authority Swissmedic (2015DR1009) and was registered at ClinicalTrials.gov (NCT02528526) before study start. All patients provided written informed consent before study participation, including consent for publication of individual anonymized data.

## Competing interests

C.N. is co-founder, director, and member of the scientific advisory board of NormOxys ®, Inc. and J.-M.L. is co-founder and chairman of scientific advisory board of NormOxys ®, Inc, this being the company that holds the patent on ITPP. M.A.S., P.L., R.G., B.H. and P.-A.C. had access to raw data. The study drug was provided by NormOxys ®, Inc, free of charge. Neither NormOxys ®, Inc nor any other funding source were involved in study design, patient recruitment/care, data collection, analysis/interpretation, or manuscript writing. The remaining authors declare no competing interests.
