## [Peer Review File · Nature Communications]

REVIEWER COMMENTS

Reviewer #1 (Remarks to the Author) (HPB oncology, trials):

To the authors:

This is an interesting topic. I am including some comments which may assist the authors in improving their manuscript.

Abstract:

Within the abstract, it states that 28 patients with advanced hepatopancreatobiliary malignancies were enrolled. However, within results, it states that 8 of these patients had colorectal cancer (colorectal liver mets), therefore, this should be clarified within the abstract.

Introduction:

Within introduction, line 101, it mentions that evofosfamide had shown promising results. References are provided, but can you please add tumour types here, to add context.

What is the evidence for use of drugs targeting hypoxia in HPB/colorectal malignancies? What was the rationale for this current trial in patients with HPB and colorectal malignancies? Please expand within introduction.

Just a minor point, the abbreviation used for "Cholangiocarcinoma" is usually "CCA" not "CCC".

Results (safety sections):

Line 151, replace ...AE were judged to be owed to the... with ...AE were judged to be related to the...

Radiological response: Line 183, can you clarify at which point response was evaluated – first scan or other?

Biochemical response: Can you clarify in line 203 which circulatory tumor markers you are referring to? All/some?

Acknowledging that this is phase Ib study, it would still be helpful to know what the median PFS and OS are for patients included.

Discussion:

Line 229: it is stated that treatment had high tolerability in the "severely ill". Please alter this statement, as inclusion criteria specified that patients had an ECOG PS \leq 1, and patients with infections etc would normally be excluded, so I would not classify these patients as "severely ill".

Line 245, it is stated that disease stabilization may be due to the natural course of the disease.

Another point to consider is that it may be related to the degree of hypoxia in tumors. Can this point be discussed, as ideally you may want to select patients for inclusion in trials of these types of agents, with above a median value for hypoxia to optimize chance of success. This should be explored a little.

Line 259, there is a typo. Should be ...as evidenced....and not ...as evinced.

Table 1:

There is a typo. Should read "Palpitations" in the relevant row.

Reviewer #2 (Remarks to the Author) (HPB oncology, trials):

This is a phase 1 of an pro-vasculature compound and has found MTD for subsequent studies. That is the ONLY output of merit. The paper is unusual and disconcerting in that in addition to this outcome:

1 Outcomes for systemic treatment that occurs after dosing are reported. It should not report "post treatment treatment" outcomes.

2 The circulating biomarkers are non-specific surrogates and are correlated with outcome. This should not be done. They really needed to get either imaging or direct tissue outcomes if they wanted to find out more as these types of biomarkers have been shown to be unhelpful in the past.

3 It does not combine the drug with chemotherapy, almost the only scenario in which other angiogenic agents have been successful.

Reviewer #3 (Remarks to the Author) (hypoxia):

Currently there are no strategies for targeting tumor hypoxia. However, many methods for targeting hypoxia and hypoxic cells have been considered. In the current manuscript, Myo-inositol trispyrophosphate (ITPP) is proposed as a novel re-oxygenating compound with no apparent toxicity in preclinical models. ITPP has been show to increase the efficacy of subsequent chemotherapy by normalizing blood vessel vasculature thereby aiding drug delivery. The current paper reveals the results of a 3+3 dose-escalation phase 1b study (NCT02528526) involving 28 patients with advanced hepatopancreatobiliary malignancies that received nine 8h-infusions of ITPP over three weeks across eight dose levels and this was followed by standard chemotherapy. The maximum tolerated dose was 12,390 per mg/m². Adverse events (32) were reported such as hypercalcemia which required little or no intervention. This is an interesting and timely study but it did not include molecular correlates to suggest that the drug indeed reduced hypoxia-target genes. This would have greatly improved the study and attempted examine HIF-expression or hypoxia-targeted genes. This would also help to inform if a lower than MTD dose may have the same efficacy.

***Phase Ib dose-escalation study of Myo-inositol trispyrophosphate, a novel hypoxia-modifier,
in patients with hepatopancreatobiliary tumors***

Response to Reviewers

Reviewer #1

Question 1: *This is an interesting topic. I am including some comments which may assist the authors in improving their manuscript. Abstract: Within the abstract, it states that 28 patients with advanced hepatopancreatobiliary malignancies were enrolled. However, within results, it states that 8 of these patients had colorectal cancer (colorectal liver mets), therefore, this should be clarified within the abstract.*

Answer 1: We thank reviewer #1 for the thorough review of our manuscript and the valuable suggestions, which helped us to improve our manuscript. We have added the term ‘*and liver metastases of colorectal cancer*’ to hepatopancreatobiliary malignancies in the abstract and the manuscript where appropriate. We have left the title unchanged due to restrictions in word numbers. If the reviewers prefer a further modification of the title, we propose to add ‘*primary and secondary*’ hepatopancreatobiliary tumors.

Q2: *Introduction: Within introduction, line 101, it mentions that evofosfamide had shown promising results. References are provided, but can you please add tumour types here, to add context.*

A2: The anti-tumor effects of evofosfamide as well as PX-473 were examined preclinically and clinically in different tumor entities, with both drugs extensively tested in colorectal and pancreatic cancers, which were also targeted in our study. Considering the reviewer’s suggestions regarding evofosfamide, we have added the following section including adequate references to the manuscript: “*Similarly, hypoxia-activated prodrugs such as evofosfamide, releasing bromoisophosphoramidate mustard in hypoxic tumor microenvironment, have shown promising results in preclinical, phase I & II studies in pancreatic, biliary, liver and colorectal cancer as well as soft tissue sarcoma among other tumor types. However, it failed to show benefits on survival in large scale phase III trials of soft tissue sarcoma and further clinical development was subsequently abandoned.*”

Q3: *What is the evidence for use of drugs targeting hypoxia in HPB/colorectal malignancies? What was the rationale for this current trial in patients with HPB and colorectal malignancies? Please expand within introduction.*

A3: The rationale for selection of HPB and colorectal malignancies is based on the following considerations:

(i) Tumor hypoxia is a crucial parameter resulting in effects on both cancer cells and associated cells of the tumor microenvironment, such as epithelial-to-mesenchymal transition with subsequent increase of metastatic potential¹, modification of innate and adaptive immunity^{2, 3}, and cellular metabolism⁴. These aspects all contribute to an elevated risk for local failure and distant metastasis⁵. In several large-scale transcriptional analyses of human tumor cohorts, HPB/colorectal malignancies show intermediate to strong hypoxia scores^{6, 7} and therefore depict ideal targets for testing

anti-hypoxic therapies. As outlined in question and answer 2, previous anti-hypoxic therapies such as evofosfamide and PX-473 were indeed tested in colorectal, pancreatic, liver and biliary cancer.

(ii) Our collaborating colleagues from the University of Strasbourg/France (research group of Prof. Jean-Marie Lehn, Nobel laureate in chemistry) developed ITPP, the first anti-hypoxic small chemical molecule being well tolerated in animals⁸. ITPP has the potential to impact on a multitude of hypoxia mediated effects, rather than current therapeutic strategies focusing on one specific aspect of tumor hypoxia (e.g. VEGF-inhibitor bevacizumab for tumor angiogenesis). Also ITPP has been extensively tested in small animal models of hypoxic gastrointestinal tumor entities during its development, providing preclinical data for murine colorectal^{9, 10, 11}, hepatoma¹² and pancreatic¹³ cancers. Based on the chemical development reports on hypoxia as well as preclinical studies, our hypothesis was that restoring normoxia in hypoxic HPB/colorectal neoplasms is beneficial especially in multimodal combination with cytotoxic agents^{10, 13}.

(iii) Lastly, our clinical and basic research laboratory focuses on the pharmaceutical and surgical treatment of gastrointestinal and hepatopancreatobiliary (HPB) diseases. We developed multiple preclinical animal models for the study of HPB^{14, 15} and colorectal malignancies¹⁶ and tested several novel pharmacological^{17, 18, 19, 20} and surgical approaches^{21, 22, 23, 24} for treatment of tumors in a translational manner in patients.

Based on the reasons outlined above, we decided for the inclusion of the selected 4 gastrointestinal tumor types. The respective modified paragraph detailing the rationale and evidence for the selection of HPB/colorectal malignancies has been removed from the methods and added to the introduction.

Q4: *Just a minor point, the abbreviation used for “Cholangiocarcinoma” is usually “CCA” not “CCC”.*

A4 This abbreviation has been corrected throughout the manuscript and figures.

Q5: *Results (safety sections): Line 151, replace ...AE were judged to be owed to the... with ...AE were judged to be related to the...*

A5: This has been corrected.

Q6: *Radiological response: Line 183, can you clarify at which point response was evaluated – first scan or other?*

A6: Thank you for this comment. To clarify that the first assessment refers to the imaging after ITPP monotherapy, we adjusted the according sentence as follows: “Of the 27 patients assessed for efficacy *after ITPP monotherapy*, 14 patients had morphologically stable disease (SD)....”.

Q7: *Biochemical response: Can you clarify in line 203 which circulatory tumor markers you are referring to? All/some?*

A7: We added the corresponding circulatory tumor marker (carcinoembryonic antigen (CEA) for CRLM, α -fetoprotein (AFP) for HCC and carbohydrate antigen 19-9 (CA19-9) for PDAC and CCA) for each tumor entity in the revised version of the manuscript.

Q8: *Acknowledging that this is phase Ib study, it would still be helpful to know what the median PFS and OS are for patients included.*

A8: As per reviewer's suggestion, we calculated median progression free survival (PFS) and overall survival (OS) and implemented in the manuscript in a new paragraph termed "survival" in the results section. However, as also stated in the question of the reviewer, readers of our manuscript need to acknowledge the phase 1b character of this study with a consecutive limited conclusion regarding efficacy and survival due to the small and heterogenous sample size. Therefore, we additionally mention these PFS and OS data in the discussion section of our manuscript to avoid overstatement.

Q9: *Discussion:Line 229: it is stated that treatment had high tolerability in the "severely ill". Please alter this statement, as inclusion criteria specified that patients had an ECOG PS \leq 1, and patients with infections etc would normally be excluded, so I would not classify these patients as "severely ill".*

A9: This term has been changed to "patients with *advanced tumor burden*".

Q10: *Line 245, it is stated that disease stabilization may be due to the natural course of the disease. Another point to consider is that it may be related to the degree of hypoxia in tumors. Can this point be discussed, as ideally you may want to select patients for inclusion in trials of these types of agents, with above a median value for hypoxia to optimize chance of success. This should be explored a little.*

A10: Indeed, this is a very important point raised by the reviewer and might substantially contribute for differences observed in response rates among different tumors and patients. In our initial study protocol, we had planned to obtain both pre- and post-treatment tumor biopsies as well as 18F-Fluoromisonidazole positron emission tomography (18F-FMISO PET) imaging for assessment of tissue tumor hypoxia. Unfortunately, the independent ethics committee reviewing our initial phase 1b clinical trial protocol objected to repeated excision of tumor biopsies due to the invasiveness of the procedure and FMISO PET due to additional radiation exposure. We were therefore not able to correlate the degree of pre-existing tumor hypoxia to response rates on ITPP treatment or changes in hypoxia mediated gene expression under anti-hypoxic treatment with ITPP.

We have adjusted the respective passages in the discussion to make the reader aware of the aspect of underlying degree of tumor hypoxia.

Q11: *Line 259, there is a typo. Should be ...as evidenced....and not ...as evinced.*

A11: This has been corrected.

Q12: *Table 1: There is a typo. Should read “Palpitations” in the relevant row.*

A12: This has been corrected.

Q13: This is a phase 1 of an pro-vasculature compound and has found MTD for subsequent studies. That is the ONLY output of merit. The paper is unusual and disconcerting in that in addition to this outcome:

A13: We thank reviewer 2 for the critical review of our manuscript. Indeed, this is a phase 1b clinical trial reporting the first-in-patient application of a novel first-in-its-class medical drug, the anti-hypoxic compound myo-Inositol-trispyrophosphate (ITPP). ITPP is an allosteric effector on hemoglobin, shifting the oxygen-dissociation curve to the right and therefore promoting hypoxia especially in hypoxic areas such as tumor tissue.

From a medical drug development point of view, phase I clinical trials are of outmost interest and always consist of the mandatory assessment of safety and tolerability with subsequent identification of a maximum tolerated dose (MTD) of novel compounds. Phase 1 studies are *per se* not intended to provide definitive statements on efficacy of compounds, as reliable conclusions on efficacy are not possible due to increasing doses and small number of patients in cohort.

By identification of the MTD and reporting all obligatory safety and pharmacodynamic data, our study has fulfilled its aim as phase 1 trial and complied with the predefined study endpoints as determined and accepted by the responsible ethics committee and published at study start on public trial registries. Additionally, our study provides first insight into anti-tumor efficacy of ITPP by assessment of both morphologic (MRI) and metabolic (18FDG-PET-CT) radiological tumor responses under ITPP treatment and subsequent chemotherapy. Furthermore, our results offer first evidence regarding potential anti-hypoxic efficacy in patients by evaluation of changes in serum tumor markers and experimental angiogenesis markers in a translational manner. For the revised version of the manuscript, we are now additionally providing tissue based markers of hypoxia of one patient, for which biopsies before and after ITPP treatment were available.

In turn, we acknowledge the phase 1b character and shortcomings of the study at hand and do not intend to overstate our findings. Findings on anti-tumor efficacy must be perceived with caution and are preliminary at best due to the small and heterogenous sample sizes. However, we think this study is of considerable interest for readers of *Nature Communications* and researchers exploring hypoxia, since this first-in-class pharmaceutical agent might be applied in various oncological settings or diseases and biological processes associated with hypoxia. This study offers the necessary safety and tolerability data for researchers to continue the clinical exploration of ITPP.

Q14: 1 Outcomes for systemic treatment that occurs after dosing are reported. It should not report "post treatment treatment" outcomes.

A14: We report outcomes after both ITPP monotherapy and subsequent chemotherapy ('post treatment treatment'). Since this is the first-in-patient application of the study drug, the study setup was defined based on preclinical safety and efficacy data. In our murine experiments, we observed best tolerability when ITPP was applied sequentially before systemic cytotoxic chemotherapy (e.g. used as a sensitizer or preconditioner, please refer also to answer 16)¹⁰.

ITPP indirectly affects tumor growth by restoration of normoxia and reversal of hypoxia-mediated molecular alterations. As shown in our preclinical experiments, effects on normalization of tumor vasculature are long lasting¹⁰. This implies that ITPP can be administered independent of chemotherapy, might provide its anti-hypoxic effects for a lasting period and therefore is amenable to subsequent chemotherapy. However, ITPP itself has no direct antitumoral activity. Our interdisciplinary study team therefore decided to report outcomes "post ITPP monotherapy" as well as "post subsequent

chemotherapy". By separating the administration of ITPP and chemotherapy, our study allowed a more concise evaluation of ITPPs direct effect on tumor growth and serum angiogenesis markers independent of the effect of chemotherapy. The current study setup also prevented any side effects which might be owed due to combined toxicity of simultaneous administration of ITPP and cytotoxic agents. For routine future clinical application, ITPP could potentially be administered simultaneous with chemotherapy.

Q15: 2 The circulating biomarkers are non-specific surrogates and are correlated with outcome. This should not be done. They really needed to get either imaging or direct tissue outcomes if they wanted to find out more as these types of biomarkers have been shown to be unhelpful in the past.

A15: We agree that systemic levels of the circulating markers of angiogenesis examined in our study can only serve as surrogate markers of actual tumor hypoxia. However, several studies have shown that hypoxia in tumors is e.g. linked to VEGF expression^{25, 26, 27} and systemic levels of VEGF^{28, 29, 30}. It can therefore be assumed that systemic levels of these biomarkers correlate to actual tumor hypoxia to a certain degree. We acknowledge the phase 1b design and small sample size and are aware that any correlations are preliminary at best.

Indeed, tumor tissue biopsies before and after ITPP treatment for assessment of hypoxia-mediated gene expression and biomarkers would have been favorable for this phase 1b clinical trial and would potentially have provided meaningful insight into the anti-hypoxic effects of ITPP, but were precluded by the responsible ethic committee due to the invasiveness of the procedure and related safety concerns in this phase I trial. The responsible independent ethics committee responsible also objected due to repeated FMISO PET due to additional radiation exposure, which could have aided to assess tumor hypoxia via imaging. This has been clarified in the revised manuscript.

However, one patient (Nr. 21) has had a surgical intervention (left hemi-hepatectomy) before ITPP administration, followed by another intervention (laparotomy & microwave ablation of liver metastases) after chemotherapy subsequent to ITPP. To answer the reviewers comments, we have analyzed these tissues and have indeed found signs of anti-hypoxic effects of ITPP depicted by decreased carbonic anhydrase 9 and glucose transporter 1 expression with cocomitant increased signs of vascular stability and integrity. Please refer to the new paragraph "Tissue marker responses on anti-hypoxic therapy in one patient" in the results section and new figure 4 in the manuscript.

For the planned phase II trial, we plan to measure tumor hypoxia more precisely using either FMISO-PET³¹ or BOLD-MRI³². We are also currently assessing the possibility to obtain tumor biopsies before and after ITPP treatment to perform detailed molecular assessments of hypoxia mediated gene and protein expression in cancer cells.

Q16: 3 It does not combine the drug with chemotherapy, almost the only scenario in which other angiogenic agents have been successful.

A16: Thank you for this important comment. In our preclinical experiments, we assessed different application schedules of combining ITPP and cytotoxic chemotherapy, including simultaneous application (ITPP and chemotherapy administered together), application of ITPP with subsequent chemotherapy delayed by several hours or days as well as sequential application (first administration of several cycles of ITPP followed later by chemotherapy)¹⁰. In our preclinical small animal models, simultaneous application of ITPP combined with standard chemotherapy was associated with significantly

increased morbidity and mortality compared to sequential application. For safety reasons, our interdisciplinary study team therefore decided upon a sequential application of the study drug followed by the the standard chemotherapy for this phase 1b study. For future studies, we plan to combine ITPP with simultaneous chemotherapy.

Q17: Currently there are no strategies for targeting tumor hypoxia. However, many methods for targeting hypoxia and hypoxic cells have been considered. In the current manuscript, Myo-inositol trispyrophosphate (ITPP) is proposed as a novel re-oxygenating compound with no apparent toxicity in preclinical models. ITTP has been shown to increase the efficacy of subsequent chemotherapy by normalizing blood vessel vasculature thereby aiding drug delivery. The current paper reveals the results of a 3+3 dose-escalation phase 1b study (NCT02528526) involving 28 patients with advanced hepatopancreatobiliary malignancies that received nine 8h-infusions of ITTP over three weeks across eight dose levels and this was followed by standard chemotherapy. The maximum tolerated dose was 12,390 per mg/m². Adverse events (32) were reported such as hypercalcemia which required little or no intervention. This is an interesting and timely study but it did not include molecular correlates to suggest that the drug indeed reduced hypoxia-target genes. This would have greatly improved the study and attempted to examine HIF-expression or hypoxia-targeted genes. This would also help to inform if a lower than MTD dose may have the same efficacy.

A17: We thank reviewer 3 for this very valuable comment. As described in Q10/A10 and Q15/A15, the absence of tumor tissue obtained before and after ITTP treatment for assessment of hypoxia-governed gene expression is a limitation of the current study. However, this phase 1b study aims at reporting safety and tolerability data rather than efficacy of a novel medical drug.

The section in our original study protocol application proposing tumor biopsies was rejected by the independent ethics committee together with the FMISO-PET (due to additional radiation exposure of participants), as repeated biopsies of tumor sites in the liver or pancreas were deemed too invasive and carrying a relevant morbidity (e.g. postinterventional bleeding or the inherent risk of tumor cell dispersion) for a first-in-patient clinical trial aiming at assessment of safety and tolerability rather than efficacy. We agree that assessment of tissue-based molecular targets of hypoxia (e.g. HIF1 α expression/nuclear translocation or transcription of downstream targets of hypoxia responsive elements) would have been very desirable and allowed to assess the anti-hypoxic effects of ITTP on tumor cells more in detail.

To overcome these limitations, we used morphologic (using MRI) and metabolic (using PET-CT) imaging to assess tumor response. Moreover, we decided to assess serum-based surrogate markers of hypoxia and tumor vasculature as the best available targets. We are aware that these markers are just surrogates and do not imply causality; however, they might mirror tumors' hypoxic response towards ITTP and the observed correlations between downregulation of angiogenic markers and improved outcomes warrant further clinical investigation in phase II trials. We therefore plan for the phase II trial to reapply for both tissue biopsies and oxygenation imaging.

References

1. Rankin EB, Giaccia AJ. Hypoxic control of metastasis. *Science* **352**, 175-180 (2016).
2. Noman MZ, *et al.* PD-L1 is a novel direct target of HIF-1 α , and its blockade under hypoxia enhanced MDSC-mediated T cell activation. *The Journal of experimental medicine* **211**, 781-790 (2014).
3. Chouaib S, Noman MZ, Kosmatopoulos K, Curran MA. Hypoxic stress: obstacles and opportunities for innovative immunotherapy of cancer. *Oncogene* **36**, 439-445 (2017).
4. Al Tameemi W, Dale TP, Al-Jumaily RMK, Forsyth NR. Hypoxia-Modified Cancer Cell Metabolism. *Front Cell Dev Biol* **7**, 4 (2019).
5. Walsh JC, Lebedev A, Aten E, Madsen K, Marciano L, Kolb HC. The clinical importance of assessing tumor hypoxia: relationship of tumor hypoxia to prognosis and therapeutic opportunities. *Antioxidants & redox signaling* **21**, 1516-1554 (2014).
6. Bhandari V, *et al.* Molecular landmarks of tumor hypoxia across cancer types. *Nat Genet* **51**, 308-318 (2019).
7. Ye Y, *et al.* Characterization of hypoxia-associated molecular features to aid hypoxia-targeted therapy. *Nature Metabolism* **1**, 431-444 (2019).
8. Duarte CD, Greferath R, Nicolau C, Lehn JM. myo-Inositol trispyrophosphate: a novel allosteric effector of hemoglobin with high permeation selectivity across the red blood cell plasma membrane. *Chembiochem* **11**, 2543-2548 (2010).
9. Limani P, *et al.* The Allosteric Hemoglobin Effector ITPP Inhibits Metastatic Colon Cancer in Mice. *Annals of surgery* **266**, 746-753 (2017).
10. Limani P, *et al.* Antihypoxic Potentiation of Standard Therapy for Experimental Colorectal Liver Metastasis through Myo-Inositol Trispyrophosphate. *Clin Cancer Res* **22**, 5887-5897 (2016).
11. Derbal-Wolfrom L, *et al.* Increasing the oxygen load by treatment with myo-inositol trispyrophosphate reduces growth of colon cancer and modulates the intestine homeobox gene Cdx2. *Oncogene* **32**, 4313-4318 (2013).
12. Aprahamian M, *et al.* Myo-InositolTrisPyroPhosphate treatment leads to HIF-1 α suppression and eradication of early hepatoma tumors in rats. *Chembiochem* **12**, 777-783 (2011).
13. Raykov Z, *et al.* Myo-inositol trispyrophosphate-mediated hypoxia reversion controls pancreatic cancer in rodents and enhances gemcitabine efficacy. *Int J Cancer* **134**, 2572-2582 (2014).
14. Lehmann K, *et al.* Liver failure after extended hepatectomy in mice is mediated by a p21-dependent barrier to liver regeneration. *Gastroenterology* **143**, 1609-1619.e1604 (2012).
15. Linecker M, *et al.* How much liver needs to be transected in ALPPS? A translational study investigating the concept of less invasiveness. *Surgery*, (2016).
16. Limani P, *et al.* Selective portal vein injection for the design of syngeneic models of liver malignancy. *Am J Physiol Gastrointest Liver Physiol* **310**, G682-688 (2016).
17. Heinrich S, *et al.* Adjuvant gemcitabine versus NEOadjuvant gemcitabine/oxaliplatin plus adjuvant gemcitabine in resectable pancreatic cancer: a randomized multicenter phase III study (NEOPAC study). *BMC Cancer* **11**, 346 (2011).

18. Tschuor C, *et al.* Constitutive androstane receptor (Car)-driven regeneration protects liver from failure following tissue loss. *Journal of hepatology*, (2016).
19. Linecker M, *et al.* Perioperative Omega-3 fatty Acids Fails to Confer Protection in Liver Surgery. Results of a multicentric, double-blind, randomized controlled trial. *Journal of hepatology*, (2019).
20. Petrowsky H, *et al.* Effects of pentoxifylline on liver regeneration: a double-blinded, randomized, controlled trial in 101 patients undergoing major liver resection. *Annals of surgery* **252**, 813-822 (2010).
21. Tschuor C, *et al.* Salvage parenchymal liver transection for patients with insufficient volume increase after portal vein occlusion -- an extension of the ALPPS approach. *Eur J Surg Oncol* **39**, 1230-1235 (2013).
22. Linecker M, *et al.* The ALPPS Risk Score: Avoiding Futile Use of ALPPS. *Annals of surgery*, (2016).
23. Petrowsky H, Fritsch R, Guckenberger M, De Oliveira ML, Dutkowski P, Clavien P-A. Modern therapeutic approaches for the treatment of malignant liver tumours. *Nature Reviews Gastroenterology & Hepatology*, (2020).
24. Petrowsky H, McCormack L, Trujillo M, Selzner M, Jochum W, Clavien PA. A prospective, randomized, controlled trial comparing intermittent portal triad clamping versus ischemic preconditioning with continuous clamping for major liver resection. *Annals of surgery* **244**, 921-928; discussion 928-930 (2006).
25. Ziemer LS, Koch CJ, Maity A, Magarelli DP, Horan AM, Evans SM. Hypoxia and VEGF mRNA expression in human tumors. *Neoplasia (New York, NY)* **3**, 500-508 (2001).
26. Cvetkovic D, *et al.* Increased hypoxia correlates with increased expression of the angiogenesis marker vascular endothelial growth factor in human prostate cancer. *Urology* **57**, 821-825 (2001).
27. Vergis R, *et al.* Intrinsic markers of tumour hypoxia and angiogenesis in localised prostate cancer and outcome of radical treatment: a retrospective analysis of two randomised radiotherapy trials and one surgical cohort study. *The Lancet Oncology* **9**, 342-351 (2008).
28. Dunst J, *et al.* Tumor hypoxia and systemic levels of vascular endothelial growth factor (VEGF) in head and neck cancers. *Strahlenther Onkol* **177**, 469-473 (2001).
29. Salven P, Mänpää H, Orpana A, Alitalo K, Joensuu H. Serum vascular endothelial growth factor is often elevated in disseminated cancer. *Clin Cancer Res* **3**, 647-651 (1997).
30. Le Q-T, Courter D. Clinical biomarkers for hypoxia targeting. *Cancer metastasis reviews* **27**, 351-362 (2008).
31. Padhani AR, Krohn KA, Lewis JS, Alber M. Imaging oxygenation of human tumours. *European radiology* **17**, 861-872 (2007).
32. Yang DM, Arai TJ, Campbell JW, 3rd, Gerberich JL, Zhou H, Mason RP. Oxygen-sensitive MRI assessment of tumor response to hypoxic gas breathing challenge. *NMR Biomed* **32**, e4101-e4101 (2019).

REVIEWER COMMENTS

Reviewer #1 (Remarks to the Author):

Comments have been adequately addressed.

Reviewer #2 (Remarks to the Author):

I thank the authors for their responses. I am afraid these have simply reiterated the submitted manuscript in more detail rather than provided any fundamental insights or rationale. Ultimately this is a 1b study with some candidate biomarker data. There is no mechanistic dissection of the mechanism with supporting translational data. As such the study is valuable but at best a report of observed phenomena.

Reviewer #3 (Remarks to the Author):

The authors were unable to address my concern due to the lack of tissue available for studies but they described this in the updated text.

Zurich, April 26th, 2021

*Phase Ib dose-escalation study of the hypoxia-modifier Myo-inositol trispyrophosphate
in patients with hepatopancreatobiliary tumors*

Response to Reviewers (2nd round of revisions)

Reviewer #1

Comments have been adequately addressed.

We thank reviewer #1 for the thorough review of our manuscript. The valuable suggestions helped us improve our manuscript.

Reviewer #2

I thank the authors for their responses. I am afraid these have simply reiterated the submitted manuscript in more detail rather than provided any fundamental insights or rationale. Ultimately this is a 1b study with some candidate biomarker data. There is no mechanistic dissection of the mechanism with supporting translational data. As such the study is valuable but at best a report of observed phenomena.

We thank reviewer #2 for the constructive criticisms and feedback. Since this is a phase Ib trial, mechanistic conclusions are statistically difficult to support, based on the ethically restricted patient number.

Reviewer #3

The authors were unable to address my concern due to the lack of tissue available for studies but they described this in the updated text.

We thank reviewer #3 for the review of our manuscript and the points raised, which have helped to improve the manuscript significantly. We have addressed all concerns and implemented the suggestions of all reviewers. As outlined in the manuscript and discussed in the first responses to reviewers, tumor tissue biopsies before and after ITPP treatment for assessment of hypoxia-mediated gene expression and biomarkers would have been favorable for this phase Ib clinical trial and would potentially have provided meaningful insight into the anti-hypoxic effects of ITPP, but were precluded by the responsible ethic committee due to the invasiveness of the procedure and related safety concerns in this phase I trial. The lack of systematic tissue analysis has been addressed and clarified in the revised manuscript. We are planning to perform tissue biopsies for the upcoming phase II study.